# Ultrafast Yb-Doped Fiber Laser Using Few Layers of PdS_2_ Saturable Absorber

**DOI:** 10.3390/nano10122441

**Published:** 2020-12-06

**Authors:** Ping Kwong Cheng, Shunxiang Liu, Safayet Ahmed, Junle Qu, Junpeng Qiao, Qiao Wen, Yuen Hong Tsang

**Affiliations:** 1Department of Applied Physics and Materials Research Center, The Hong Kong Polytechnic University, Hung Hom, Kowloon, Hong Kong, China; 18078019r@connect.polyu.hk (P.K.C.); safayetahmed.s@gmail.com (S.A.); qiao68910@163.com (J.Q.); 2Key Laboratory of Optoelectronic Devices and Systems of Ministry of Education and Guangdong Province, College of Optoelectronic Engineering, Shenzhen University, Shenzhen 518060, China; 2150190124@email.szu.edu.cn (S.L.); jlqu@szu.edu.cn (J.Q.)

**Keywords:** PdS_2_, group 10 TMDs, ultrafast laser, AFM, TEM, SEM, XPS

## Abstract

Two-dimensional (2D) transition metal dichalcogenide (TMD) materials have exceptional optoelectronic and structural properties, which allow them to be utilized in several significant applications in energy, catalyst, and high-performance optoelectronic devices. Among other properties, the nonlinear optical properties are gaining much attention in the research field. In this work, a unique pentagonal TMD material, palladium disulfide (PdS_2_), is employed as a saturable absorber (SA) in an ytterbium-doped fiber (YDF) laser cavity and mode-locked laser pulse is generated. At first, liquid phase exfoliation is performed to prepare PdS_2_ nanoflakes. Afterward, the PdS_2_-nanoflakes solution was incorporated in the side-polished fiber (SPF) to form SPF-based PdS_2_-SA. By utilizing this SA, a highly stable mode-locked laser pulse is realized at pump power of 160 mW, which has a center wavelength of 1033 nm and a 3-dB spectral bandwidth of 3.7 nm. Moreover, the pulse duration, maximum power output and corresponding single-pulse energy were determined as 375 ps, 15.7 mW and 0.64 nJ, respectively. During the experiment, the mode-locked pulse remained stable till the pump power reached a value of 400 mW and, for the regulation of power, the slope efficiency is calculated at about 4.99%. These results indicate that PdS_2_ material is a promising nonlinear optical material for ultrafast optical applications in the near-infrared (NIR) region.

## 1. Introduction

Ultrafast fiber lasers produce short to ultrashort laser pulses, which can be utilized in various advanced applications such as surface structural technology, laser cutting [1,2], eye surgery [3], lunar laser communication [4], 3D city mapping [5,6], optical information processing and nonlinear optics [7,8]. Generally, ultrafast fiber lasers can be generated by utilizing Q-switching or mode-locking mechanisms with a pulse duration ranging from microseconds to femtoseconds. Compared to active laser pulse generator devices such as acousto-optic modulator (AOM) and electro-optic modulator (EOM), passive modulator saturable absorbers (SA) are cost-effective, simple, small, and easy to fabricate [9,10,11]. Therefore, nowadays, the study of SAs is becoming a significant research focus in laser technology and laser physics. 

Recently, various kinds of nonlinear materials such as graphene, black phosphorous (BP), topological insulators (TIs), carbon nanotubes and transition metal dichalcogenides (TMDs) [12,13] were studied as high-performance SA materials with low loss due to their admirable saturable absorption properties and ultrafast response time [14]. Among these materials, two dimensional (2D) TMDs provide a lot of advantages over other SA materials owing to their superior features such as tunable bandgap [15,16,17], strong nonlinear optical properties [18], ultrafast carrier dynamics, wide operation spectrum [19], and a short carrier valley relaxation time [20]. The generation of both Q-switching and mode-locking laser pulses was studied earlier using these nonlinear TMDs materials. For example, layered WS_2_ and MoS_2_ were employed in different hybrid AOM solid-state laser systems for compressing the pulse duration of the Q-switched pulse [21,22]. In addition, the mode-locked pulse was also achieved by utilizing a TMD-based SA (WSe_2_ [23], MoSe_2_ [24]). Meanwhile, to the best of our knowledge, in the 1 μm ytterbium-doped fiber (YDF) laser cavity, only eight different types of layered TMD materials (including our previous result) were successfully used as SAs for the generation of mode-locking pulses in the near-infrared (NIR) region [25,26,27,28,29,30,31]. Therefore, new research is ongoing to find new TMDs materials that can be used in a 1-µm YDF laser cavity for achieving ultrashort laser pulses. 

Lately, a novel group 10 TMDs material, palladium disulfide (PdS_2_) is gaining attention in the scientific field owing to its unique lattice arrangement. Generally, most of the layered TMDs material is structured like a hexagonal lattice, where one layer of transition metal atoms (from group 4 to group 10) is sandwiched between two layers of chalcogen atoms (S, Se, and Te). However, as shown in Figure 1, PdS_2_ is shaped as an interesting pentagonal structure that facilitates excellent electronic and optical tunability characteristics distinct from the conventional hexagonal structure [32,33,34]. Additionally, layered PdS_2_ shows a semi-metallic property when it is formed between bulk and bilayer (BL) forms and the electronic structure of monolayer (ML) PdS_2_ has an indirect bandgap with a 1.1 eV bandgap value [35]. The literature has demonstrated various material properties of layered PdS_2_ material on the photonic and photoelectric fields. For instance, Wang et al. presented the photoluminescence nature of PdS_2_ quantum dots (QDs) [36]. Additionally, Saraf et al. reported the photocatalytic property of ML PdS_2_, which was proposed for utilization in water-splitting applications for hydrogen and oxygen evolutions [37]. Furthermore, the multilayered PdS_2_ was deposited on a side-polished fiber (SPF) as a saturable absorber for femtosecond ultrafast laser generation in the erbium-doped fiber cavity. Meanwhile, the saturable absorption properties of PdS_2_ based SPF-SA were tested by using a 2 ps ultrafast laser source at 1564 nm central wavelength and 1.7% modulation depth, and 0.24 GW/cm^2^ saturable intensity in transverse electric mode, were observed [13]. These findings indicate that the layered PdS_2_ material has promising photonic applications and can be a suitable candidate for near-infrared mode-locking pulse generation. 

In this paper, it is demonstrated that, by incorporating na SPF-based PdS_2_ SA into a YDF laser cavity, highly stable mode-locked laser pulses can be achieved. The mode-locked YDF laser was realized when the pump power was regulated between 160 and 400 mW. The repetition rate and the pulse duration were measured to be 24.4 MHz and 375 ps, respectively, at the pump power of 160 mW. Our findings noted that Pd-based TMDs possess potential for mode-locked fiber laser applications. To the best of our knowledge, this is the first time that palladium disulfide (PdS_2_) SAs are being used for the generation of mode-locking laser pulses in 1.03-µm ytterbium-doped fiber laser cavity. Therefore, this work will open up a new research area for PdS_2_ materials in a near-infrared region. 

## 2. Fabrication and Characterization of PdS_2_

At first, 0.175 mg of PdS_2_ powder (6N, Six Carbon Inc., Shenzhen, China) is sonicated at 400 W ultrasonic power and 40 kHz of frequency for 25 h after mixing it with 175 mL of Isopropyl alcohol solvent (IPA). Then, the as-sonicated solution is centrifuged at 3000 rpm for 7 min to ensure a reduction in the non-exfoliated particles and bulk impurities from the solution. The PdS_2_-IPA supernatant is then spin-coated on the silicon substrate and dried on the hot plate at 60 °C to avoid the accumulation of PdS_2_ flakes together. 

The topological characteristic of PdS_2_ flakes is tested by using atomic force microscopy (AFM, Bruker Nanoscope 8, Billerica, MA, USA). After analyzing the 220 as-prepared flakes of PdS_2_, the average lateral dimension of the long axis, short axis, and thickness are determined as 182, 114, and 44 nm, respectively, as shown in Figure 2a–c. Meanwhile, two height profiles of randomly selected flakes from the corresponding AFM image are presented in Figure 2d. The lateral dimension along the marked line of Flake A and Flake B is around 100 and 210 nm, as observed from the inset of Figure 2d. Moreover, the heights of those two selected flakes are determined as about 46 nm. From these results, the topologic properties of the exfoliated PdS_2_ particles can be determined. 

Furthermore, the as-exfoliated PdS_2_ sample was observed by using field emission transmission electron microscopy (FETEM, JEOL JEM-2100F, Tokyo, Japan) and Scanning Electron Microscope (SEM, Tescan VEGA3, Brno, Czech Republic). An image of randomly selected PdS_2_ flake from the TEM image is provided in Figure 3a. The grain size of this flake is around 110–190 nm, which is in agreement with the AFM result. In addition, from the high-resolution TEM image (Figure 3b), the crystallinity of this PdS_2_ flake is also observed. In the HRTEM image, four crystal lattice planes are presented, which are denoted as (311), (113), (020), and (111), with d-spacing of 1.71, 2.32, 2.80, and 3.57 Å, respectively. Meanwhile, the corresponding selected area electron diffraction (SAED) patterns of this PdS_2_ flake are presented in Figure 3c. Five polymorphic rings are clearly observed, which consist of four previously found crystal lattice planes of (311), (113), (020), and (111), and one additional lattice plane of (220) with a d-spacing of 1.95 Å [38]. In Figure 3d, the energy-dispersive X-ray spectroscopy (EDS) spectrum of PdS_2_ samples obtained from SEM measurement is shown. From the EDS spectrum, it can be observed that there are three types of signals (Au, Pd, and S) present. The signal of Au is recorded due to the utilization of Au coating during the measurement. Therefore, it can be verified that the sample has no impurities. 

The chemical composition of raw PdS_2_ powder was tested by employing X-ray photoelectron spectroscopy (XPS, Thermo Fisher Scientific ESCALAB 250Xi, Waltham, MA, USA) measurement with Al Kα X-ray source. Figure 4a presents the Pd 3d spectrum, with two obvious peaks of 3d_5/2_ and 3d_3/2_, which are located at the binding energies of 336.2 and 341.6 eV. Meanwhile, the well-fitting result of the S 2p spectrum is observed with two doubles, as shown in Figure 4b, which can be caused by the peak splitting of PdS_2_ [39,40]. The 2p_3/2_ and 2p_1/2_ of S(I) state are found at 161.2 and 163.6 eV binding energy, respectively. While the 2p_3/2_ and 2p_1/2_ of S(II) state (fitting at 162.2 and 164.6 eV, respectively) have a similar spectrum to 2p_3/2_ and 2p_1/2_ of S(I) state. These results are well matched with the previous study of PdS_2_ material [25]. The atomic ratio of Pd and S is about 1:2.23 and agrees with the stoichiometric ratio of the PdS_2_ molecular structure. In addition, there are no obvious sulfate signals, which reveals that the PdS_2_ sample has high stability at ambient.

## 3. Ultrafast Photonics Applications

In Figure 5, The schematic of the YDF laser cavity is provided. A 980-nm laser diode with the maximum power of 700 mW was used to pump the YDF laser (Nufern, Yb SM-YSF-HI-6/125, East Granby, CT, USA) via a wavelength-division multiplexer. The fiber resonator was composed of a 1 m YDF with 250 dB/m absorption at 980 nm. The polarization independent isolator (PI-ISO) (Nufern, PI-ISO, East Granby, CT, USA) was used to force the unidirectional signal propagation inside the cavity while the signal was coupled out using 10% optical coupler (OC). A polarization controller (PC) was used to facilitate the initialization of the mode-lock operation by doing a fine-tune of the intra-cavity polarization. Here, a D-shaped fiber (DF) was immersed in PdS_2_ material to fabricate the SA. The interaction length of the D-shaped area was 5 mm, and the distance from the fiber core boundary to the D-shaped area was 1 μm. The total cavity length was 8.53 m, consisting of a 1.0 m YDF and ~7.53 m single-mode fiber and the total dispersion of our cavity was estimated to be 0.45 ps^2^.

In this experiment, by using PdS_2_-based SA, the generation of stable continuous wave mode-locked (CWML) laser pulses is demonstrated, and the optical performances are measured by utilizing a 5 GHz photodetector (THORLABS, DET08CFC, Newton, NJ, USA), a 4 GHz oscilloscope (LECROY, WaveRunner 8404M, New York, NY, USA), an optical spectrum analyzer (Yokogawa, AQ6370C, Tokyo, Japan), a frequency analyzer (RIGOL, DSA815, Beijing, China), and an optical power meter (THORLABS, PM20CH, Newton, NJ, USA). These results are displayed in Figure 6a–d. A CWML regime is achieved when the pump power is larger than 160 mW, corresponding to the output power of 3.25 mW. The mode-locked pulse train is presented in Figure 6a. The time interval between two pulses is 40.9 ns, which corresponds to a 24.4 MHz repetition rate. From the inset of Figure 6a, the uniform intensity of the pulse train can be observed, which indicates the high stability of the laser operation. From Figure 6b, it can be determined that the optical spectrum of mode-locked pulses is centered at 1033 nm and the 3-dB spectral width is 3.7 nm. From the radio frequency (RF) spectrum, shown in Figure 6c, a strong signal peak with a fundamental repetition rate of 24.4 MHz can be clearly observed. Moreover, the signal to noise ratio (SNR) value is measured at about 65 dB. From the inset of Figure 6c, the RF spectrum for 500 MHz range can be determined, which indicates the high stability of the obtained laser operation. The measured autocorrelation trace of the corresponding mode-locked pulse is shown in Figure 6d, with a pulse duration of about 375 ps. Additionally, the time bandwidth product (TBP) is calculated as 390, which implies that the mode-locked pulsed is heavily chirped. From Figure 6e, an excellent linear relationship between average output power and pump power with a slope efficiency of 4.99% can be determined. The output power curve illustrated that the laser started to operate in a continuous wave (CW) regime when the pump power increased to 95 mW, and as the pump power further increases to the threshold value 160 mW, stable continuous wave mode lock (CWML) is achieved. Finally, by observing the pulse spectrum for over 8 h of laser operation (Figure 6f), it can be verified that there is no signal alteration, and the stability of the achieved mode-locked pulse is very high. From Table 1, it can be determined that the obtained laser performance is comparable with other TMDs-SA based mode-locked lasers operating in YDF laser cavity. The achieved output power is the highest and the achieved pulse duration is the narrowest among all other TMDs SA-based results, except for NiS_2_. Moreover, with this SA it is possible to obtain one of the highest optical-to-optical conversion efficiency for YB-doped mode-locked lasers. Overall, it can be observed that the archived experimental results are better compared with other TMDs-SA. These comparisons suggest that PdS_2_-SA can be an extraordinary nonlinear material to generate ultrafast pulses laser with high output power. Hence, layered PdS_2_ is an excellent functional material as a saturable absorber for a 1-µm laser cavity system.

## 4. Conclusions

In this study, the generation of a Yb-doped mode-locked fiber laser pulse with a central wavelength of 1033 nm has been experimentally realized by utilizing SPF-based PdS_2_-SA. The pulse laser performance includes a 3-dB spectral bandwidth of 3.7 nm, a fundamental repetition of 24.4 MHz and pulse duration of 375 ps. Moreover, the high stability of the generated laser pulse is determined by observing the wave spectrum while keeping the pump power at 160 mW. There is no noticed alternation of the spectrum. Therefore, these results indicate that the PdS_2_-SA has the potential to develop novel photonic and optoelectronic devices in the near-infrared region.

## Figures and Tables

**Figure 1 nanomaterials-10-02441-f001:**
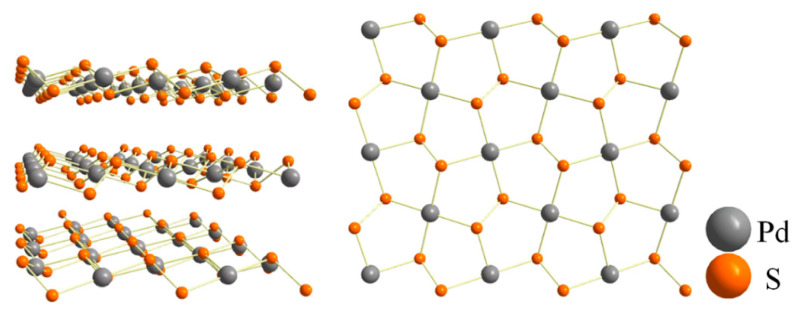
Atomic structure of 2D layered PdS_2_.

**Figure 2 nanomaterials-10-02441-f002:**
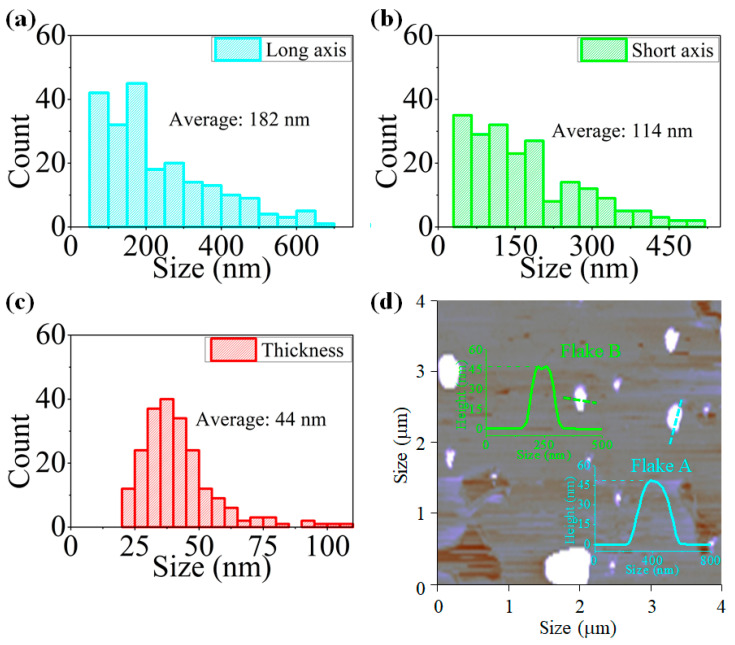
Statistical AFM measurement of 220 PdS_2_ flakes, the lateral dimension of (**a**) long axis, (**b**) short axis and (**c**) thickness. (**d**) The image and height profile of selected Flake A and B along the marked line.

**Figure 3 nanomaterials-10-02441-f003:**
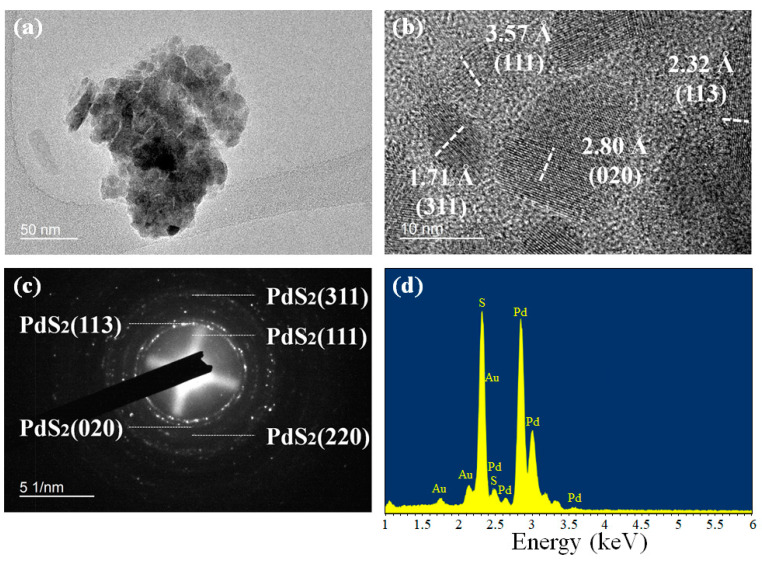
(**a**) An image of randomly selected flake with (**b**) high-resolution TEM image, and (**c**) corresponding SAED pattern by FETEM measurement. (**d**) EDS spectrum by SEM measurement.

**Figure 4 nanomaterials-10-02441-f004:**
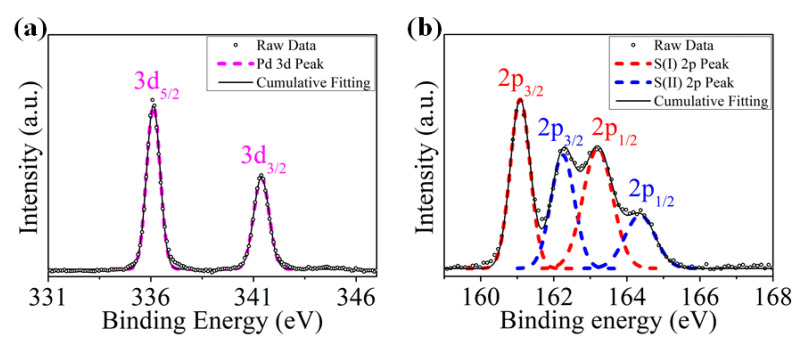
PdS_2_ XPS spectrum ranging from (**a**) Pd 3d and (**b**) S 2p regions.code.

**Figure 5 nanomaterials-10-02441-f005:**
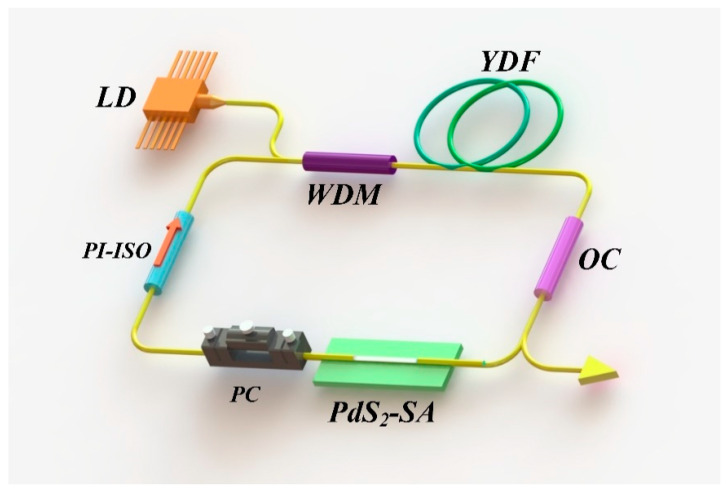
Schematic illustration of passively mode-locked YDF lasers based on PdS_2_ SA. LD: laser diode. WDM: wavelength division multiplexer. YDF: ytterbium-doped fiber. OC: output coupler. PdS_2_-SA: PdS_2_ saturable absorber. PC: polarization controller. PI-ISO: polarization independent isolator.

**Figure 6 nanomaterials-10-02441-f006:**
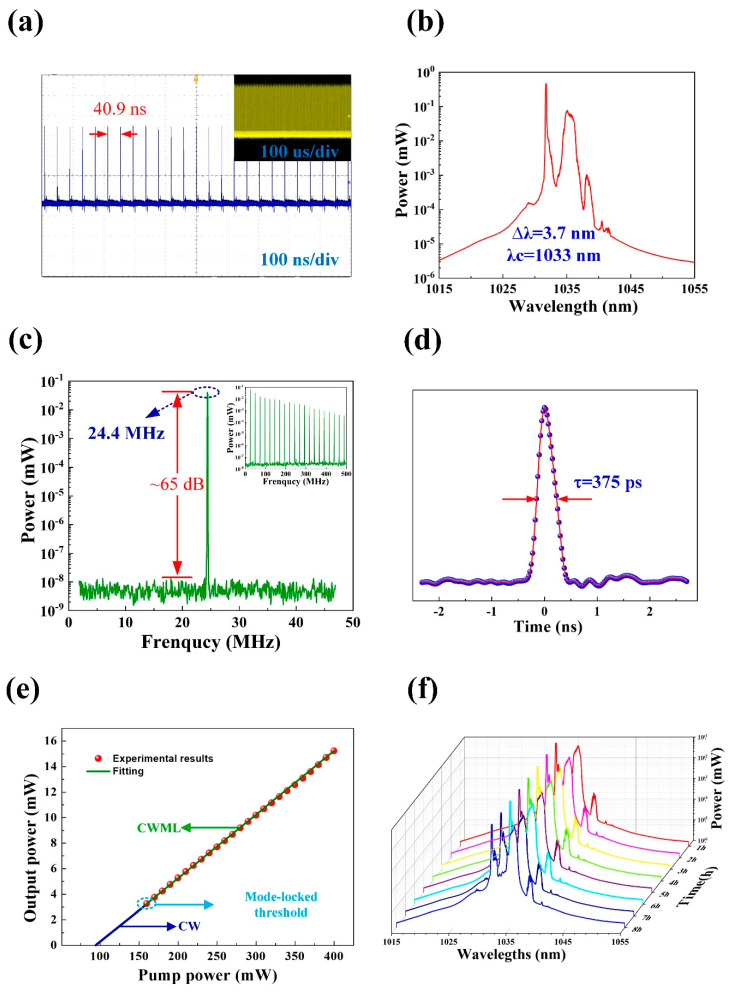
Typical mode-locked pulse characteristics. (**a**) Pulse train. (**b**) Optical spectrum. (**c**) Radio frequency spectrum (inset: the wideband RF spectrum) of the mode-locked pulses. (**d**) measurement of the laser pulse width. (**e**) Variation of the output power with respect to the pump power. (**f**) Optical spectra measurements at 1h intervals over 8 h of operation.

**Table 1 nanomaterials-10-02441-t001:** Various TMD-based SAs for mode-locking operation in Ytterbium-doped fiber laser system.

Group of Transition Metal	Material	Wavelength, nm (3 dB Bandwidth, nm)	Pulse Duration (SNR, dB)	Slope Efficiency (Output Power, mW)	Reference
Group 5	NbSe_2_	1033 (0.155)	380 ps (43)	3.70% (10.5)	[25]
Group 6	MoS_2_	1042.6 (8.6)	656 ps (59)	1.1% (2.37)	[26]
	MoSe_2_	1040 (4.26)	471 ps (54)	/ (2.0)	[27]
	WS_2_	1030.3 (1.1)	2.5 ns (48)	2.5% (8.02)	[28]
Group 10	NiS_2_	1064.5 (7.8)	11.7 ps (66)	5.1% (35.6)	[29]
	PtSe_2_	1064 (2.0)	470 ps (53)	3.6% (12.19)	[30]
	PdSe_2_	1067.4 (5.22)	768 ps (61)	4.6% (15.6)	[31]
	PdS_2_	1033 (3.7)	375 ps (65)	4.99% (15.7)	This work

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
