# Peer review of "Ultrafast Yb-Doped Fiber Laser Using Few Layers of PdS2 Saturable Absorber"

_nanomaterials, 2020, doi:10.3390/nano10122441_

Round 1

Reviewer 1 Report

Report on “Ultrafast Yb-doped fiber laser using few-layers group

 10 materials PdS2 as a saturable absorber” Chang et. al.

In this contribution  the authors reported ultrafast 1 um fiber laser using PdS2 materials  as a saturable absorber. This paper can be consider for publication after major revision.

In the literature there is large number of papers devoted to ultrafast fiber lasers using a passive saturable absorber. The authors should clearly defined why the PdS2 is better that previously materials used for passive  mode-locking.

In Figure 6 b,c,d,f  y axis is described as an intensity however is should  be rather described as power. Power is give in dBm unit not intensity.

Why the pulses are so long, in the range of 375 ps?

What need to be done to obtain femtosecond pulse duration?

Moreover in my opinion this paper also required carefully English proofreading.

Author Response

Comment: Report on “Ultrafast Yb-doped fiber laser using few-layers group 10 materials PdS2 as a saturable absorber” Chang et. al.

In this contribution the authors reported ultrafast 1 um fiber laser using PdS2 materials as a saturable absorber. This paper can be consider for publication after major revision.

Response:  We thank the reviewer for his valuable response. His comments will definitely help us to improve our manuscript.

Question 1: In the literature there is large number of papers devoted to ultrafast fiber lasers using a passive saturable absorber. The authors should clearly defined why the PdS2 is better that previously materials used for passive mode-locking.

Answer 1: Thank you reviewer for your comment. In this paper, we have demonstrated the mode locking pulse laser generation by using PdS2 based saturable absorber. Utilizing this SA, it possible to achieve a pulse duration of 375 ps and a maximum output power of 15.7 mW. Here, the achieved pulse width is the second narrowest pulse width and the realized power also is the second highest output power attained compare to other TMDs based SA in Yb doped fiber cavity, as shown in table 1. (page 9).

Also, the optical performance has great stability, which can be proven by the large signal-to-noise ratio (SNR) of about 65 dB. And the long-term stability of the PdS2 based mode locking pulse generation is confirmed by observing the output spectrum for over 8 hours of operation as shown in figure 6f. (page8)

Furthermore, the nonlinear optical property of PdS2 material was also experimental demonstrated at Er-doped fiber laser cavity in the previous work, which represented that PdS2 material has a broad-spectrum range for pulse laser generation. (page 2, line 65-69)

These results demonstrate that PdS2 based SA has better performances than almost TMDs material-based SA.

Question 2: In Figure 6 b,c,d,f  y axis is described as an intensity however is should be rather described as power. Power is give in dBm unit not intensity.

Answer 2: Thank you for your comment. The y axis of Figure 6 b,c,d,f were changed. The y axis name of Figure 6 b, c, f are changed to Power with a milliwatt (mW). And the y axis of Figure 6d is deleted same as the pulse train of Figure 6a. (page 8)

Question 3: Why the pulses are so long, in the range of 375 ps?

Answer 3: Thank you for your comment. As we all know, single-mode fiber exhibits anomalous dispersion at 1.5 mm and the total dispersion of erbium-doped laser cavity is a negative value. The soliton is obtained under the combined action of dispersion and nonlinear effects. In an erbium-doped laser, the obtained pulse width is generally in the order of femtoseconds. On the other hand, the single-mode fiber shows normal dispersion at 1 mm and the total dispersion of the ytterbium-doped laser cavity is a positive value. As in the Yb-doped laser cavity, the dispersion compensation element is being added, therefore, the pulse width obtained is generally in the order of picoseconds. Our results are comparable to the previously published reports [1-6].

Question 4: What need to be done to obtain femtosecond pulse duration?

Answer 4: Thank you for your comment. Our laser cavity is simple without the dispersion compensation element, the pulse width is generally on the order of picoseconds. It is similar to other reports. If we want to obtain femtosecond pulses, we can add dispersion compensation elements in the laser cavity or use gratings to compress the pulses outside the cavity.

Question 5: Moreover in my opinion this paper also required carefully English proofreading.

Answer 5: Thank you for your comment. We have completely revised the English writing in the manuscript, which is yellow highlighted in the marked manuscript.

Ref:

  1. Tunable and switchable multi-wavelength dissipative soliton generation in a graphene oxide mode-locked Yb-doped fiber laser
  2. Single-walled carbon nanotube solution-based saturable absorbers for mode-locked fiber laser
  3. Mode-locked ytterbium-doped all-fiber lasers based on few-layer black phosphorus saturable absorbers
  4. Molybdenum disulfide (MoS2) as a broadband saturable absorber for ultra-fast photonics
  5. Broadband Nonlinear Photonics in Few-Layer MXene Ti3C2Tx (T = F, O, or OH)
  6. Zero-Dimensional MXene-Based Optical Devices for Ultrafast and Ultranarrow Photonics Applications

Reviewer 2 Report

The authors proposed a passive modulator saturable absorber based on nanoscale PdS2 flakes for the generation of mode-locking laser pulses in λ = 1.03 μm ytterbium doped fiber laser. They developed the technology for the production of absorbers and carried out a detailed study of the new material PdS2. Using the developed passive shutter, stable generation of ultra short pulses with duration of 375 ps at a wavelength of 1.03 μm was obtained in an ytterbium doped fiber-optic ring laser. The paper is of interest to specialists and researchers in the development and application of laser technology. May be published after minor revision.

Notes:

Lines 2,3

“Ultrafast Yb-doped fiber laser using few-layers group 10 materials PdS2 as a saturable absorber”    “group 10 materials” - delete

Replace PdS2 with PdS2

Line 109 "Therefore, it can be verified that the sample has no impurities." It doesn't work that way. The guaranteed level of zero impurities must be specified.

Author Response

Comments and Suggestions for Authors

The authors proposed a passive modulator saturable absorber based on nanoscale PdS2 flakes for the generation of mode-locking laser pulses in λ = 1.03 μm ytterbium doped fiber laser. They developed the technology for the production of absorbers and carried out a detailed study of the new material PdS2. Using the developed passive shutter, stable generation of ultra short pulses with duration of 375 ps at a wavelength of 1.03 μm was obtained in an ytterbium doped fiber-optic ring laser. The paper is of interest to specialists and researchers in the development and application of laser technology. May be published after minor revision.

Response:  We thank the reviewer for his valuable response. His comments will definitely help us to improve our manuscript.

Notes:

Question 1: Lines 2,3 “Ultrafast Yb-doped fiber laser using few-layers group 10 materials PdS2 as a saturable absorber”  “group 10 materials” – delete Replace PdS2 with PdS2

Answer 1: Thank you for your comment. The title was changed in the new version. The new title is as follows-

Ultrafast Yb-doped fiber laser using few-layers of PdS2 saturable absorber

Question 2: Line 109 "Therefore, it can be verified that the sample has no impurities." It doesn't work that way. The guaranteed level of zero impurities must be specified.

Answer 2: Thank you for your comment. There were no other elements were detected in the EDS spectrum, while the Au signal came from the Au coating during the energy-dispersive X-ray spectroscopy measurement. Therefore, we claimed that the sample has no impurities.

Round 2

Reviewer 1 Report

After correction the paper can be accepted for publication